# Identification of Transcriptional Regulators of Immune Evasion Across Cancers: An Alternative Immunotherapeutic Strategy for Cholangiocarcinoma

**DOI:** 10.3390/cancers16244197

**Published:** 2024-12-17

**Authors:** Simran Venkatraman, Brinda Balasubramanian, Pornparn Kongpracha, Supaporn Yangngam, Nisa Chuangchot, Suparada Khanaruksombat, Suyanee Thongchot, Monthira Suntiparpluacha, Kyaw Zwar Myint, Sunhapas Soodvilai, Tavan Janvilisri, Siwanon Jirawatnotai, Peti Thuwajit, Chanitra Thuwajit, Jarek Meller, Somchai Chutipongtanate, Rutaiwan Tohtong

**Affiliations:** 1Department of Biochemistry, Faculty of Science, Mahidol University, Bangkok 10400, Thailand; venkask@ucmail.uc.edu (S.V.); brinda.balasubramanian@nottingham.ac.uk (B.B.); kyawzwarmyint@gmail.com (K.Z.M.); tavan.jan@mahidol.ac.th (T.J.); 2Department of Environmental and Public Health Sciences, University of Cincinnati College of Medicine, Cincinnati, OH 45267, USA; mellerj@ucmail.uc.edu; 3Translational Medical Sciences Unit, School of Medicine, Biodiscovery Institute, University of Nottingham, Nottingham NG7 2RD, UK; 4Quantitative Biosciences Institute (QBI), University of California San Francisco, San Francisco, CA 94158, USA; jo.kongpracha@ucsf.edu; 5Gladstone Institute of Data Science and Biotechnology, J. David Gladstone Institutes, San Francisco, CA 94158, USA; 6Department of Cellular and Molecular Pharmacology, University of California San Francisco, San Francisco, CA 94158, USA; 7Department of Immunology, Faculty of Medicine Siriraj Hospital, Mahidol University, Bangkok 10700, Thailand; yangngam@bric.ku.dk (S.Y.); nisachuang@gmail.com (N.C.); suparada.kha@mahidol.ac.th (S.K.); suyanee.thongchot@gmail.com (S.T.); peti.thu@mahidol.ac.th (P.T.); chanitra.thu@mahidol.ac.th (C.T.); 8Siriraj Center of Research Excellence for Cancer Immunotherapy, Research Department, Faculty of Medicine Siriraj Hospital, Mahidol University, Bangkok 10700, Thailand; 9Siriraj Center of Research Excellence for Precision Medicine and Systems Pharmacology, Department of Pharmacology, Faculty of Medicine Siriraj Hospital, Mahidol University, Bangkok 10700, Thailand; monthira.sun@mahidol.edu (M.S.); siwanon.jir@mahidol.ac.th (S.J.); 10Department of Physiology, Faculty of Science, Mahidol University, Bangkok 10400, Thailand; sunhapas.soo@mahidol.ac.th

**Keywords:** cancer, transcriptomics, immunotherapy, cholangiocarcinoma, immune checkpoints

## Abstract

Current immunotherapies are ineffectively designed toward single-target inhibition in a complex multifactored process. Moreover, available therapeutics are incapable of meeting the needs of the expanding cancer patient population due to resistance, recurrence, immune-related adverse events, or patients that do not express the intended target. This study proposes Master Regulators of Immune Evasion (MR-IE) inhibition as a more effective systemic therapeutic strategy that targets multiple immune evasion molecules to reinstate immune recognition and the elimination of tumors. Moreover, this study also identifies and ranks ideal candidate MR-IEs for each cancer type associated with patient mortality. This study has been validated in cholangiocarcinoma. This investigation offers MR-IE as a new avenue of therapeutic targets that can be validated and developed as an immunotherapeutic strategy for all cancer types. Moreover, we address that not all cancers would respond to the same target, and thus, we propose mortality-associated MR-IEs per cancer type.

## 1. Introduction

Cancer is a multifaceted disease that confers resilience and survival advantages over normal cells. These advantages include sustained proliferative signaling, resistance to growth suppressors, invasion and metastasis, and immune evasion [1]. During cancer development, cancer cells interact intimately with the immune cells in the microenvironment to facilitate its survival and progression. These interactions promote an immunosuppressive environment to allow cancer cells to proliferate without surveillance and elimination by immune cells. This is mediated by the expression of co-inhibitory immune checkpoints (ICs), the infiltration of pro-tumoral immune cells, the lowering of the expression of self-antigens, and the release of immune suppressive cytokines [2].

Deciphering complex mechanisms that support immune evasion has given rise to various immunotherapies that unveil cancer cells and reinstate immune recognition and elimination of cancer cells. There are various mechanisms tackled by immunotherapies, including the inhibition of co-inhibitory IC interactions by immune checkpoint blockade (ICB), cancer-cell surface target recognition, induced cytotoxicity by chimeric antigen receptor T-cell (CAR-T) therapy, and recruiting anti-tumoral immune cell infiltration by promoting inflammation using cytokine-based therapies. Of these therapies, ICBs have shown great promise in clinical use; however, only 30–40% of the patient population are eligible for these therapies, and up to 12.5% of eligible patients respond to treatment [3]. The reported reasons for the lack of response to immunotherapies primarily include acquired resistance, low expression of intended targets amongst patient populations, and even relapses after treatment [4,5]. As the ineligible patient population expands, the exploration of alternative therapeutic strategies and targets is warranted.

To efficiently impede a multifaceted process, a therapeutic target that regulates multiple elements of a pathway would be ideal. This is mitigated by transcription factors, which centrally govern multiple signaling cascades by regulating gene expression. Our previous review surveyed transcriptional regulators of ICs and oncoimmunology, so-called master regulators of immune evasion (MR-IE) [6]. Here, we describe how transcription factors, such as MYC and STAT3, play a pivotal role in the context of oncoimmunology, including the regulation of IC molecules gene expression, maturation of dendritic cells, the release of pro-tumoral cytokines, inhibiting anti-tumor immune cells, and promoting angiogenesis [7,8]. Hence, we surmise that these MR-IE would be more effective immunotherapeutic targets for cancers.

The present investigation identified (mortality-associated) MR-IE per cancer type by using robust integration bioinformatics tools to stratify and subset The Cancer Genome Atlas (TCGA), the Pan-Cancer dataset based on their IC molecule expression, and their predicted immune cell infiltration levels. We validated the top-ranking candidate in cholangiocarcinoma (CCA), as it has previously been reported to have a complex immune architecture [9], and concerning incidence and mortality rate, with limited treatment options in Thailand [10]. This investigation not only aims to introduce a novel and effective immunotherapeutic strategy by targeting MR-IE, but also offers more potent immunotherapeutic treatment options for CCA.

## 2. Methodology

### 2.1. Dataset Acquisition and Classification

TCGA Pan-Cancer Atlas RNA-Seq dataset was obtained from https://gdc.cancer.gov/about-data/publications/pancanatlas (accessed on 16 May 2020) along with its respective sample annotation, tumor subtype annotation, and its clinical data. This dataset was used to draw immune cellular fraction estimates using CIBERSORTx (https://cibersortx.stanford.edu), as described in Thorsson et al.’s investigation [11,12]. Using the datasets provided in the paper, we selected TCGA samples that were in the first and third quartiles of the CD8+ T-cell infiltration (less than 25% and greater than 75%) to represent low infiltration and high CD8+ T-cell infiltration, respectively. Additionally, we assumed the infiltration scores of CD4+ and CD8+ T-cells for the TCGA tumors and selected samples that were in the first and third quartiles to represent low and high CD4+ and CD8+ T-cell infiltration. Lastly, using the Leukocyte Fraction scores, we used the samples in the first and third quartiles to represent low and high leukocyte infiltration. Thus, using these criteria, the Pan-Cancer dataset was subset into 6 datasets (Low and High Leukocyte Infiltration, Low and High CD4+ and CD8+ T cell infiltration, and Low and High CD8+ T cell infiltration).

### 2.2. Hierarchical Clustering of TCGA Samples

The 7 datasets (6 subsets and the Pan-Cancer dataset) were transformed using log2(x + 2) to remove NA values. Using the R package pheatmap (version 1.0.12), the datasets were plotted on a heatmap that was subjected to hierarchical clustering [13]. These heatmaps were used to determine expression levels (low, intermediate, and high) of co-inhibitory IC. Samples that belonged to each cluster were extrapolated and pooled into their respective groups for subsequent analyses.

### 2.3. Differential Gene Expression Analysis

Differential gene expression analysis was conducted using the R packages edgeR (version 3.42.4) and limma (version 3.56.2) [13]. TCGA samples within low, intermediate, and high co-inhibitory IC were compared in the following directions: low vs. intermediate and low vs. high co-inhibitory IC. The *p*-values were adjusted using the Benjamani–Hochberg method. The *p*-value threshold was set below 0.001 (*p* < 0.001)

### 2.4. Pathway Enrichment Analyses

To assess the pathways the differentially expressed genes (DEGs) belong to, we performed enrichment analysis using the tools iLINCS Pathway Analysis (https://www.ilincs.org) [14]. The input for these tools was either the DEGs list with its fold change and adjusted *p*-values or just the gene list, wherever appropriate.

### 2.5. Transcriptional Regulators Identification

DEGs list was applied to the Search Tool for the Retrieval of Interacting Genes/Protein Database (STRING DB) (https://string-db.org) to generate a gene network. This network was used in Expression2Kinases (http://www.maayanlab.net/X2K/, accessed on 26 October 2020) to enrich transcription factors using known protein interactions to construct a network.

### 2.6. Survival Analysis

Survival analysis was performed using the survival, survminer, and survplot packages in R version 4.0.2. The Cox-Proportional Univariate Regression model was applied to each significantly enriched transcription factor against the survival data of each TCGA cancer type. This script was adapted from Mikhail Dozmorov’s analysis script found in (https://github.com/mdozmorov/TCGAsurvival?tab=readme-ov-file#readme, accessed on 17 November 2020). The Hazard Ratios (HR) and the Wald Statistic *p*-value were tabulated.

### 2.7. Composite Scoring of Transcription Factors

The composite score was calculated by multiplying the average enrichment score provided by Expression2Kinases for each significantly enriched (*p*-value < 0.05) transcription factor and the Hazard Ratio for each transcription factor against each cancer type. This was used to rank the top MR-IE candidates per cancer type.

### 2.8. Cell Lines

Cell lines KKU-M213 and RBE were purchased from the Japanese Collection of Research Bioresources Cell Bank and were maintained in a RPMI culture medium supplemented with 10% FBS (Gibco, Life Technologies, Grand Island, NY, USA) and incubated at 37 °C with 5% CO_2_.

### 2.9. Lipofectamine Transfection for MYC Knockdown

The siRNA of MYC (human) (sc-29226) was purchased from Santa Cruz (Santa Cruz Biotechnology, Dallas, TX, USA). The lipofectamine reagent was prepared using Optimem for 3 reactions. Each reaction was combined with its respective siRNA: siRNA of MYC, negative control siRNA, and reagent-control siRNA. These reactions were then applied to cells seeded on a 6-well plate and incubated for 24 h. After this occurred, the treatment medium was replaced with the culture medium.

### 2.10. Real-Time PCR (qPCR)

Total RNA was extracted using the Geneaid Total RNA extraction kit in accordance with the manufacturer’s protocol. In total, 500 ng RNA was subjected to reverse transcriptase using the Hyperscript RT mastermix kit (GeneAll, GeneAll Biotechnology, Seoul, Republic of Korea) using a Bio-Rad Thermal cycler. The cDNA was subjected to qPCR using the Agilent Real-Time PCR thermal cycler.

### 2.11. Western Blot

In total, 8 × 10^5^ cells were seeded onto a 60 mm dish for 24 h and treated with the lipofectamine reagent carrying the siRNA of interest or inhibitor treatment. The cells (after 48 h) were lysed according to the protocol defined in the publication [15]. Protein samples of 40 μg were loaded and separated using 12% SDS-PAGE and transferred to a nitrocellulose membrane. The membrane was blocked with a 5% bovine serum albumin (BSA) and probed with primary antibodies at 1:1000 dilution. After incubation, the membrane was washed for 10 min in TBS/T, 3 times, incubated with secondary antibodies conjugated with horseradish-peroxidase (HRP), and immunoreactive bands were visualized by the ECL Plus Western Blotting Detection System (Bio-Rad Laboratories, Inc., Hercules, CA, USA) using G:Box ChemiXL 1.4 (Syngene, Synoptics, Cambridge, UK). Densitometry analysis was performed using Image Lab^TM^ software version 6.1.

### 2.12. Proteomics

To investigate the protein expression profile of cancer cells upon inhibiting the candidate MR-IE, the extracted proteins were prepared according to the protocol listed in a previous investigation [16]. Cells were lysed using a 1% Triton-X buffer supplemented with a protease inhibitor cocktail. The resulting extracted proteins were precipitated using acetone stored at −20 °C for 16 h. After precipitation, the protein pellet was reconstituted in 0.25% RapidGest SF (Waters, Cheshire, UK) in 10 mM ammonium bicarbonate. The peptides were subjected to LC-MS/MS. The spectrum data were collected using an Orbitrap HF hybrid mass spectrometer. The raw data files were analyzed using MaxQuant (https://www.maxquant.org) to obtain protein identifications and their respective label-free quantification values using in-house standard parameters. Normalization and differential protein expression analyses were performed on the relative protein abundance matrix conducted using the Bioconductor package ‘DEP’ (version 1.22.0) in R. The cutoff for candidates considered significantly differentially expressed was Log2Fold change in log2(2) at a *p*-value < 0.05. The results were visualized using the ‘Enhanced Volcano’ (version 1.18.0) and ‘ggplot2’ packages (https://ggplot2.tidyverse.org). Differentially expressed proteins were then analyzed using iLINCS Signaling Pathway Impact Analysis (SPIA) while the KEGG pathway analysis database was used for signaling pathway enrichment.

### 2.13. Cell Viability and Proliferation Assays

In total, 5 × 10^3^ cells, per well, were seeded in a 96-well plate for 24 h, and treated with either a vehicle control or an MYC inhibitor (10074-G5, purchased from MedChemExpress, Monmouth Junction, NJ, USA). Cell viability was assessed at 24 h intervals, up to 72 h post-treatment. The vehicle control was a 0.1% DMSO-containing complete medium. MTT assay was performed according to the standard protocol [15]. For assays that require 96 h treatment, 1 × 10^3^ cells were seeded in a 96-well plate for 24 h and then treated.

### 2.14. Isolation of Peripheral Blood Mononuclear Cells (PBMCs) and Activation of T-Cells

Blood from 50 mL of healthy volunteers was drawn, and PBMCs were isolated by the lymphocyte separating medium (Corning, Mediatech Inc., Manassas, VA, USA), and centrifuged according to density. A total of 5 × 10^6^ cells/well was then seeded in a 12-well plate in an AIM-V cell culture medium (Gibco, Life Technologies, Grand Island, NY, USA) and incubated for 2 h at 37 °C for monocyte cell adhesion. The resultant cell suspensions are non-adherent T-cells, which were cultured in an AIM-V cell culture medium (Gibco, Life Technologies, Grand Island, NY, USA) in the presence of phytohemagglutinin and cytokines IL-2 for 3 days.

### 2.15. Production of Fourth Generation FRα-CAR T-Cells

The fourth generation anti-FRα-CAR-T cells have been established by Luangwattananun et al., who indicated these cells for the treatment of breast cancer cell lines. The characterization and details of their construct have been established in their publication [17]. Briefly, the construct contains a single specific-chain variable fragment targeting FRα. In the fourth-generation iteration of this CAR-T cell, three intracellular costimulatory domains (CD28, 4-1BB, and CD27) linked to CD3ζ were added to the construct. Our investigation repurposed this CAR-T cell toward CCA cells expressing FRα. The isolated T-cells were transduced with FRα-CAR lentiviruses at a multiplicity of infection (MOI) of 50 using a reported method [17]. T-cells were cultured in an AIM-V cell culture medium containing a 5% human AB serum, as well as IL-2, IL-7, and IL-15 for 4 days. The expression of FRα-CAR T-cells was assayed by using the flow cytometry (BD Biosciences, Becton Dickson and Company, Franklin Lakes, NJ, USA) technique after the cells stained with mouse anti-FRα antibodies (1:100 in 1% BSA/PBS, Thermo Fisher Scientific, MA, USA) and goat anti-mouse IgG (1:100, Thermo Fisher Scientific, Waltham, MA, USA), and Alexa Fluor^®^488 (1:500, Thermo Fisher Scientific) were applied.

### 2.16. Immune-Mediated Cell Death Assay

The cholangiocarcinoma cell line, KKU-M213, was stained with red fluorescence protein (mCherry), and RBE was stained with CellTracker^TM^ Orange (Thermo Fisher Scientific, MA, USA). These cancer cells were seeded onto a 96-well plate (Corning, Mediatech Inc., VA, USA) at a cell density of 5 × 10^3^ cells/well. After 24 h, the cells were treated with varying concentrations of the MYC inhibitor (10074-G5), then incubated for 48 h. These cells were co-cultured with the FRα-CAR T-cells at a ratio of 2.5:1 for 24 h, after which red fluorescence intensities were measured for KKU-M213. The cell viability of RBE was measured using Crystal Violet Staining. RBE cells were fixed with methanol and stained with 0.2% Crystal violet for 20 min. The stains were washed with PBS and eluted with 0.5% SDS diluted in PBS.

Cancer cell death was estimated from reduced fluorescence intensity or absorbance values when compared to respective untreated controls. These were visualized using GraphPad Prism 9.3 as a box-and-whisker plot. Along with the fluorescent intensity and absorbance measurements, cell morphology was observed using the ECLIPSE Ti (Nikon Instruments Inc., Melville, NY, USA) microscope.

### 2.17. Statistics and Data Analysis

Data were collected in triplicates and three independent rounds, wherever applicable. Student *t*-test or one-way ANOVA with Tukey’s post hoc test was conducted to assess statistical significance at *p* < 0.05 (*).

## 3. Results

### 3.1. Analytical Pipeline

Immune evasion occurs through the synchronous process of immune cell infiltration, co-inhibitory IC, and the modulation of immune-related markers. Thus, an ideal therapeutic target would be a transcriptional regulator that governs all three processes. To identify transcriptional regulators that control these processes, we must obtain all the genes that are affected in varying degrees of estimated immune cell infiltration, and varying degrees of inhibitory IC. To do so, the Pan-Cancer TCGA samples must be subset according to varying degrees of estimated infiltration and then stratified into varying degrees of co-inhibitory IC. The transcriptome of these data subsets and clusters are then compared with each other to unravel the different signaling pathways associated with the immune evasion process. As the resulting DEGs reflect all genes associated with the process of immune evasion, the DEGs from these comparisons are used as an input for transcription factor enrichment. This analysis helps to identify transcription factors that bind common motifs identified among the DEGs. Thus, this yields candidate MR-IEs from varying degrees of leukocyte estimates and IC across cancers. We acknowledge that not all cancers are driven by the same targets, in fact, some therapeutic targets may have opposite roles in different cancers. To overcome this issue, we employed survival analysis to assess the effect of the gene expression of these transcription factors on patients’ overall survival per cancer type. Using the Hazard Ratio provided from this analysis, and the enrichment score of the transcription factor enrichment analysis, a composite score was developed to rank these candidate MR-IEs per cancer type. Together, this bioinformatics workflow identifies and ranks transcriptional regulators of immune evasion that affect genes involved in varying degrees of immune infiltration and varying degrees of IC per cancer type. This workflow is summarized in Figure 1.

### 3.2. TCGA Samples Can Be Stratified Based on Estimated Immune Cell Infiltration and Immune Checkpoint Expression 

In our investigation, we employed a deconvolution algorithm, CIBERSORTx, to estimate the proportion of immune cells in TCGA bulk-tissue transcriptomic data. Subsequently, we utilized unsupervised hierarchical clustering to further stratify the samples into clusters of low, intermediate, and high IC (Appendix A, Figure 2 and Appendix A). Interestingly, varying degrees of leukocyte estimation resulted in the formation of different clusters of samples of low, intermediate, and high IC gene expression levels (Appendix A). Particularly, CD4+ and CD8+ T-cell estimates resulted in more distinct clusters of mainly low and high co-inhibitory IC, with a few samples having an intermediate expression of IC molecules (Figure 2a,b). Thus, the presence of these cell types, in particular, may influence the expression of co-inhibitory ICs in cancer cells.

### 3.3. Immune Evasion Is Driven by Various Signaling Cascades Depending on Different Levels of Immune Checkpoint Expression and Leukocyte Estimation

To look at all the genes associated with the immune evasive process, we compared the transcriptome differences between low, intermediate and high ICs across different immune cell estimates (Appendix A). When looking at transcriptome level changes between low and high IC across the varying subsets of leukocyte estimates, we unraveled the signaling pathway changes that are associated with the immune evasion phenomenon. High or intermediate immune checkpoint-expressing samples represent an immunologically cold (anti-tumoral immune cells do not surveil) tumor, whereas low immune checkpoint-expressing samples represent immunologically hot (anti-tumoral immune cells recognize and eliminate cancer cells) tumors.

Accordingly, our differential gene expression analysis between the differing levels of IC reveals an inhibition of the Natural Killer (NK) cell-mediated cytotoxicity pathway and the T-cell receptor signaling pathway among clusters of intermediate and high IC (Figure 2d and Appendix A). Notably, in subsets of data that were highly estimated with leukocytes (Appendix A), antigen processing and presentation pathways inhibited between clusters with intermediate to high IC. Moreover, cytokine–cytokine receptor signaling pathways were activated, and interestingly, chemokine signaling pathways were inhibited in these estimated subsets of high CD4+ and CD8+ T-cells (Appendix A). This suggests that the levels of IC, particularly those in the infiltration of CD4+ and CD8+ T-cells, may trigger the secretion of cytokines to promote an immunosuppressive environment. Interestingly, amongst the low immune-cell estimated subsets (Figure 2d and Appendix A), the inhibition of NK-cell-mediated cytotoxicity and the T-cell receptor signaling pathways were maintained. However, several cancer-specific pathways were inhibited, such as cAMP signaling and cGMP-PKG signaling.

Collectively, our robust workflow incorporating estimated immune infiltration of tumors and IC stratifications of tumors confirmed and unraveled pathways related to immune evasion. This provides confidence that the DEGs would be able to identify a transcriptional regulator central to these pathways.

### 3.4. EGR1 and MYC Are Identified as Top-Ranking MR-IE in Multiple Cancers

Using the DEGs obtained from the aforementioned comparisons across differentially estimated immune cell infiltration and different IC levels, we sought to identify common transcriptional regulators that bind to the common motifs among these DEGs using Expression2Kinases. The transcription factor enrichment analysis resulted in 21 statistically significant candidate MR-IEs across the various subsets and IC clusters. To assess the likelihood of these candidate MR-IEs in regulating the expression of ICs, we assessed the co-expression between the transcription factors and ICs across all cancers. This revealed groups of candidate MR-IEs that are positively correlated with specific ICs and negatively correlated with other ICs (Figure 2e). For example, the candidate MR-IE, FOXA2, is significantly positively correlated with the expression of ARG1, FGL1, PVR, PVRL2, CEACAM, LGALS3, HMGB1, and PD-L1 (CD276). However, it is negatively correlated with the expression of VEGFB and TGFB1. This provides insights into the potential positive or negative regulation of these candidate MR-IEs of the ICs. Nevertheless, there may be lineage-specific co-expression patterns that were not accounted for in this analysis.

The identified transcriptional regulators across all cancer types are statistically scored. However, different transcriptional regulators may have varying effects on patient mortality. Thus, we conducted a cox-proportionate univariate regression analysis of each transcriptional regulator for each of the 33 TCGA cancer types to yield a hazard ratio. Using this hazard ratio multiplied by the enrichment score allowed us to rank the significant transcription factors per cancer type. The hazard ratio, enrichment scores, and the combined composite score of candidate MR-IE per cancer type are tabulated in Appendix A. The top-ranking candidate MR-IE per cancer type is depicted in Figure 3, Appendix A. The cancer types are listed as per the TCGA study abbreviations.

Notably, EGR1 appeared as the top-ranking candidate MR-IE in multiple cancer types, such as GBM, DLBC, ESCA, LUAD, LUSC, MESO, ACC, STAD, COAD-READ, BLCA, PAAD, OV, UCEC, and CESC, Whereas MYC appears to be the top-ranking candidate particularly for liver and renal associated cancers (i.e., LIHC, CHOL, KICH, KIPAN, KIRC, and KIRP). Nevertheless, MYC was frequently ranked second in cancer types, with EGR1 as the top-ranking candidate. Thus, these two candidate MR-IEs, ranked based on mortality association and statistical significance, would make ideal immunotherapeutic targets. Other candidate MR-IEs include TP53 for BRCA, AR for THCA, FLI1 for TCGT, UCS and UVEM, and PPARD for PRAD.

Our investigation focuses on providing therapeutic solutions for a pressing issue in Thailand, cholangiocarcinoma (CCA; CHOL). CCA is a prevalent malignancy with a complex tumor microenvironment. Unfortunately, current therapeutic options are incapable of treating complex heterogeneous patient populations. Thus, this investigation assesses MYC as an immunotherapeutic target for CCA.

### 3.5. MYC Perturbation in CCA Results in Downregulation of PD-L1

To validate MYC as an MR-IE for CCA, we assessed its role in regulating the expression of a representative IC molecule, PD-L1. In KKU-M213 cells, we observe that a therapeutic inhibition of MYC using a MYC inhibitor (10074-G5) (Appendix A) resulted in a significant downregulation of protein and gene expression of PD-L1 (Figure 4a,b). These results were maintained when the MYC gene was knocked down using siRNA interference (Figure 4d and Appendix A). Contrastingly, the RBE cells displayed a modest decrease in protein expression but no significant difference in gene expression (Figure 4e–h). Several factors could have contributed to the discrepancy in the molecular regulation of PD-L1 between these cell lines, including heterogeneity within CCA and unexplored compensatory mechanisms. Thus, in this investigation, we further explored KKU-M213 as the hypothesized ‘responsive’ cell line and the representative cell line for a patient population with similar molecular profiles, for which this novel immunotherapeutic strategy is surmised to benefit.

### 3.6. MYC Influences the Immune-Related Proteome in CCA Cells

We further assessed the modulation in the cytosolic immune-related proteome when MYC was perturbed using either a small-molecule or siRNA transfection (Figure 5 and Appendix A). We expected that the perturbation of the MR-IE would result in the upregulation of stimulatory immune-related markers and the downregulation of pro-tumoral immune-inhibitory markers. Notably, the perturbation of MYC had a significant effect on the proteome as ‘treated’ KKU-M213 cells, clustered distinctly from the ‘control’ samples, with a variance of 40.8% (41.1% in siMYC treated samples) between clusters (Figure 5a and Appendix A). In the MYC inhibitor-treated KKU-M213 samples, we identified 2034 proteins, of which 67 proteins were significantly differentially expressed between the treatment and control (Figure 5b,c, Appendix A). These proteins are involved in known MYC-regulated pathways, including apoptosis and cell-cycle regulation. Additionally, inflammatory pathways, such as the IL-9 signaling pathway, and interleukin signaling pathways, were also enriched (Figure 5d). The proteins specifically contributing to these pathways’ enrichment include MAP2K1, FABP5, IL-18, STMN2, and TIMM13 (Figure 5e). Remarkably, pro-tumoral markers, such as CX3CL-1 and IL-18, were down-regulated, and anti-tumoral and self-antigen markers, such as IL-18, IL-16, IL-11, HLA-A, TAP1, and TAP2, were upregulated after MYC inhibition (Appendix A).

When MYC was perturbed using siRNA transfection, we identified 2254 proteins, of which 84 were differentially expressed (Appendix A, Appendix A). These proteins were, too, enriched for inflammatory pathways, including interleukin and interferon signaling. The consistent finding provides confidence that MYC does indeed influence the immune-related proteome within CCA cells (Appendix A). Upon a closer look into the key immune-related protein expression that was modulated after treatments, we noted that the inflammation-related proteins, antigen presentation-related proteins, and TGF-beta signaling, notably, proteins related to ICs, were also differentially modulated after MYC knockdown, including SERPINB1, SERPINA4, KIR2DL5B:KIR2DL5A, and LGALS3BP (Appendix A). Consistent with the MYC inhibition treatment, pro-tumoral markers, such as IFNA-21 and IL-4R, were downregulated, whereas anti-tumoral markers, including IL-18, TAP1, and TAP2, were upregulated after the siMYC treatment (Appendix A).

Taken together, these results suggest that MYC inhibition or knockdown results in the upregulation of key effectors of inflammation via interleukins and interferons MAP2K1 and STAT1. While MYC knockdown allowed us to explore several signaling pathways, the upregulation of stimulatory immune microenvironment markers was achieved by using both methods of inhibition. The ability of MYC inhibition and knockdown to modulate various elements that contribute to the immune microenvironment shows promise for MYC inhibition to be developed into more complex models for CCA.

### 3.7. Inhibition of MYC Potentiates CAR-T-Mediated Cell Death of CCA Cells

MYC, the top-ranking MR-IE candidate of CCA, modulated multiple inhibitory elements that contribute toward immune evasion (Figure 5). Thus, inhibiting MYC in cancer cells is expected to potentiate immune-mediated recognition and elimination. Here, we inhibited MYC in CCA cells (KKU-M213 and RBE) using a small-molecule inhibitor before subjecting them to CAR-T cells to assess any cytotoxic effect the inhibitor may have on the cancer cells itself, and to determine non-lethal concentration ranges. The cancer cell lines are responsive to the MYC inhibitor treatment dose and time, dependently (Figure 6a). However, the IC_50_ of the MYC inhibitor against CCA cell viability was higher than the reported plasma concentration (58.2 µM) at 24 h and 48 h, ranging between 117.3 µM and 50.17 µM, respectively, for the KKU-M213 cell line, and 260.3 µM and 65.75 µM for the RBE cell line (Figure 6, Appendix A). Of the two cell lines, KKU-M213 was more responsive. Nevertheless, this investigation surmises that the inhibition of MYC would potentiate immune-mediated elimination and, therefore, a lower concentration could potentially elicit this effect.

To functionally assess if targeting the MR-IE of CCA reinstates immune recognition and the elimination of cancer cells, we used CAR-T cells as a model for cancer cell-targeted, immune cell-induced cytotoxicity. Acknowledging the success of fourth-generation anti-FRα CAR-T cell development in breast cancer [17], we sought to reposition this CAR-T cell toward CCA. To confirm whether this CAR-T model can be repositioned toward CCA, we assessed the expression of FRα across CCA cell lines (Appendix A). Notably, most cell lines, except HuCCA-1 and KKU-100, displayed between 20% and 70% expression, especially in KKU-M213 (~30%) and RBE (~55%), the two cell lines used in this investigation which showed similar levels of expression. (Appendix A). Specifically, the relative mean fluorescent intensity (rMFI) within these cells (except HuCCA-1 and KKU-100) ranges between two and five. Hence, the anti-FRα CAR-T cell has the potential to be repositioned toward the treatment of CCA.

To highlight that KKU-M213 is the responsive cell line and representative of the potentially responsive CCA patient population, we assessed if MYC inhibition could potentiate immune-mediated cell death. To do so, we compared the effect of MYC inhibition on target cells (CCA cells) alone versus target cells co-cultured with anti-FRα CAR-T cells. Interestingly, the effect of the MYC inhibitor was more pronounced when KKU-M213 cells were co-cultured with anti-FRα CAR-T cells, as immune-mediated cell death was potentiated (Figure 6b). Remarkably, the concentration required to elicit a 50% reduction in cell viability is much lower than when KKU-M213 cells were treated alone (Figure 6c). More notably, the effect of KKU-213 cytotoxicity was remarkably lower than MYC inhibition treatment alone and anti-FRα CAR-T cell therapy alone. This suggests that there is a synergism between the combination of two treatments. Conversely, in RBE, the CCA cell line, was deemed ‘resistant’ because it had no significant changes in IC markers after MYC inhibition or knockdown, which did not potentiate immune-mediated cell death. While RBE cells were responsive to MYC inhibition-induced cytotoxicity, co-culture with CAR-T cells did not increase cell death. This suggests that the cell death observed may be a direct effect of the MYC inhibitor on the cancer cells alone (Appendix A).

Cumulatively, this shows that MYC inhibition exhibits immunomodulatory effects that sensitize KKU-M213 cells to CAR-T-mediated cell death. Thus, MR-IE inhibition has the potential to be an effective therapeutic strategy in CCA patients represented by KKU-M213 cells. While this result is promising, complex models that assess the infiltration of immune cells, should be utilized to further develop MYC inhibition as an immunotherapeutic strategy. Hence, this should encourage the further development of MYC inhibition as an immunotherapeutic strategy for CCA, and by extension, MR-IE inhibition as an effective immunotherapeutic strategy.

## 4. Discussion

Immune evasion is a meticulous process that synchronizes several signaling cascades within tumor cells and their surrounding stromal cells to facilitate the proliferation and invasion of tumors to secondary sites [18]. These include the expression of co-inhibitory IC molecules on tumor cells and immune cells, the infiltration of pro-tumoral immune cells, downregulated self-antigen presentation, and the secretion of immunosuppressive cytokines [19]. Thus, impeding any one pathway leaves room for several mechanisms to compensate for inhibition. Existing cancer immunotherapies improved patient outcomes using ICB in lung [20], renal [21], bladder [22], and head-and-neck cancer [23]. However, only 30–40% of patients are eligible or respond to these treatments [6]. Hence, this warrants the need for more potent and context-specific therapeutic strategies to cater to different patient populations. The present investigation offers a novel and potentially more effective therapeutic strategy by exploiting the multifunctional and multitargeted roles of transcriptional regulators of immune evasion. Our investigation identified MR-IE for each cancer type by integrating deconvolution algorithms, hierarchical clustering, transcription factor enrichment analyses, and survival analyses (Figure 1, Figure 2 and Figure 3). Therefore, this investigation provides a high-confidence list of MR-IEs as candidate immunotherapeutic biomarkers for each cancer type (Appendix A, Figure 3), to encourage the development of small-molecule inhibitors against these targets.

Tumors possess a complex architecture comprising various stromal cells, cancer-associated fibroblasts, and various immune cells. The infiltration of these cells is closely associated with clinical implications [24]. Hence, it is imperative to address the infiltration of immune cells in the tumors of the TCGA cohort, to fulfill our quest for the MR-IE. One caveat of using the TCGA Pan-Cancer dataset is the information loss of tumor-infiltrated cells and surrounding cells due to bulk-tumor sequencing. To circumvent this issue, computational methods have been developed to estimate the proportion of various immune cells by assessing the similarity of the gene signatures of samples to the gene signatures of immune cells [25]. In our investigation, we adopted the CIBERSORTx algorithm to estimate immune cell infiltration. Consistent with prior reviews, our investigation shows that high infiltration of CD8+ and CD4+ T-cells in the tumors influences inhibitory IC, and is associated with cytokine-receptor signaling pathways across cancers (Appendix A).

CCA is a heterogenous lethal malignancy of the bile duct, with its peak incidence and mortality rate in northeastern Thailand, and a growing incidence rate worldwide [26]. The lack of early-stage symptoms, diagnostic markers, and rapid progression contributes to rising mortality rates. This trickles down to a lack of effective therapeutic strategies for advanced-stage patients [10,27]. Recently, immunotherapy, including ICB, agonist antibodies, and CAR-T cell therapy, has gained traction in the treatment of CCA. However, as with other cancer types, a minority of CCA patients qualify for immunotherapy based on biomarker expression and tumor mutational burden [28]. The current objective response rate for clinical trials of ICBs, such as pembrolizumab (KEYNOTE-158; KEYNOTE-028), nivolumab, and ipilimumab (NCT02923934), ranges between 5 and 20%. Thus, to amplify therapeutic options for CCA patients, this investigation proposes the therapeutic inhibition of MYC (the top-ranking MR-IE of CCA; Figure 3). Toward this, we showed that MYC inhibition potentiated immune-mediated cell death when co-cultured with anti-FRα CAR-T cells. Thus, this investigation highlights the potential of combining MR-IE inhibition with CAR-T cell therapy for a more potent treatment outcome (Figure 6).

Our investigation reports, for the first time, that EGR1 is the most frequently high-ranked and clinically relevant MR-IE among the multiple cancer types. This was followed by MYC being the second-most frequently top-ranked MR-IE (Figure 3, Appendix A). Currently, little is known of the role of EGR1 in anti-cancer immunity, which presents an opportunity for us to explore this as an immunotherapeutic target in the stipulated cancer types. This is supported by several investigations that report EGR1 as a “master regulator” of inflammatory enhancers in macrophages, by regulating the differentiation of monocytes to macrophages, and a myriad of inflammatory genes [29,30]. This shows that our workflow has merit, in being able to aptly identify MR-IEs in the right context. Moreover, as reports seldom explore the therapeutic potential of targeting EGR1 thoroughly, prospects of the current investigation should explore this phenomenon.

MYC, on the other hand, is a well-established oncogene known to regulate several intrinsic tumor-cell mechanisms, to promote the survival, growth, and proliferation of tumor cells. Recent reviews highlight the association of MYC with the expression of PD-L1 and CD-47 [19], the regulation of T-cell activation and function, and the polarization of macrophages [31]. Consequently, MYC has become an ideal therapeutic target as an oncogene and as an immunotherapeutic target. However, the role of MYC is context-dependent, as the function of MYC is dependent on post-translational modifications and mutational burdens of its gene targets, and the interaction of the cancer cells with a tumor microenvironment [32]. For instance, MYC inactivation induced proliferative arrest in hematological cancers [33], whereas MYC inactivation in osteosarcoma resulted in terminal differentiation into bone [34]. Thus, the role of MYC and its association with patient mortality is dependent on the cancer lineage [32]. Our workflow acknowledges the context-dependent roles of transcriptional regulators by accounting for their effect on patient overall survival when ranking the candidate MR-IEs (Appendix A, Figure 3). Therefore, this ranking system yields potential MR-IEs with higher expression results in lower patient survival, thereby making them ideal therapeutic targets in the specified cancer type.

We hypothesized that the targeting of MR-IEs would reinstate immune recognition and the elimination of cancer cells by the downregulation of inhibitory ICs and the upregulation of stimulatory immune-related markers. Accordingly, our validation assays confirm the role of MYC in regulating the expression of a co-inhibitory IC molecule, PD-L1 (Figure 4). Moreover, anti-tumoral inflammatory markers, such as IL-18, IL-10, and MAP2K1, were upregulated after MYC inhibition (Figure 5) in CCA cells. The modulation of these markers is associated with the reinstatement of immune recognition of cancer cells. Most importantly, the modulation of these markers culminates in the functional reinstatement of immune-mediated cytotoxicity of CCA cells (Figure 6). To further the translatability of our novel immunotherapeutic concept, we sought to combine our therapeutic strategy with CAR-T therapy. Various clinical investigations revealed that CAR-T therapies had seen success in hematological malignancies, but were less efficacious in solid tumors due to their lack of penetrability in tumors [35]. This gave rise to the combination of CAR-T therapies with ICB to sensitize solid cancer cells to CAR-T infiltration and further potentiate immune-mediated cell death, such as pembrolizumab and mesothelin-specific CAR-T cells [36]. In a similar vein, our investigation considered the immune cell infiltration and IC in the identification of MR-IE candidates, with the aim of sensitizing cancer cells to multiple facets of immune surveillance. Additionally, we combined cancer cell inhibition of MR-IE with fourth-generation CAR-T cells in CCA cells. We observed that there is a synergistic effect of CCA cytotoxicity when compared to MYC inhibition alone and CAR-T cell therapy alone. This potentially suggests that both treatments address the limitations of the other while being more efficacious in mitigating cancer cells. Prospective studies may investigate whether the combined treatment increases immune cell infiltration of tumors and anti-tumoral inflammation in vivo. Taken together, our preclinical investigation provides confidence in our workflow to identify and rank candidate MR-IEs per cancer type that can be developed as immunotherapeutic targets.

While our robust workflow accounted for multiple aspects of immune evasion, there are limitations to our findings. Firstly, a caveat of the TCGA Pan-Cancer Atlas is varied sample size per cancer type, resulting in biases in the clinical significance of predicted MR-IE in each cancer type. Moreover, we used bulk-tissue transcriptomics to infer immune cell estimates. Alternatively, the incorporation of single-cell RNA sequencing data may provide better resolution into the interaction between highly and lowly infiltrated tumors, context-specifically. Additionally, these data do not account for heterogeneity within cancer subtypes. This was evidenced in the discrepancy in response to MR-IE inhibition between KKU-M213 and RBE cell lines. This may be due to differences in post-translational modifications of MYC’s gene targets or varying genetic alterations between the two chosen cell lines. Thus, our workflow may need to be revised to account for these variables. Our investigation was limited to the use of fewer cell line representatives. However, the cell lines chosen were based on prior reports of KKU-213 being immunogenic and responsive to cell-based immunotherapies [37,38], and RBE was seldom reported to be immunogenic. This encourages the development of more immunogenic CCA cell lines to further explore various immunotherapeutic strategies. The present investigation also attempted to explore the use of siRNA knockdown as a form of therapy; however, this did not yield any significant changes in the cell viability of CCA cells alone nor with co-culture. While the knockdown of MYC was transient enough to observe transcriptional regulation, the duration was not sufficient to last during co-culture with CAR-T therapy. Additionally, the transient knockdown may eventually lead to the compensation of protein production, thus nullifying its efficacy in CCA cytotoxicity. Thus, this investigation utilized the knockdown via siRNA treatment to highlight modulations of key markers. Lastly, further investigation within CCA is required to confirm the therapeutic efficacy of MR-IE inhibition, in more complex tumor models.

## 5. Conclusions

Our investigation collectively introduces transcriptional regulators as alternative immunotherapeutic targets for the development of cancer treatments. We also provided insights into the transcriptomic changes associated with immune evasion and its regulation. Moreover, our investigation provides a robust bioinformatics workflow that ranks and identifies MR-IE per cancer type. As a result, MYC was identified as the candidate MR-IE for CCA, that downregulates key immune-related markers such as PD-L1 expression, IFNA-21, HLA-B, and IL-18. These modulations resulted in CCA cells being sensitized to CAR-T-mediated cell death. Conclusively, we encourage the development of several small-molecule inhibitors against MYC for the treatment of CCA. We hope the findings of this investigation inspire further development of these MR-IEs as therapeutic targets to benefit a wider patient population.

## Figures and Tables

**Figure 1 cancers-16-04197-f001:**
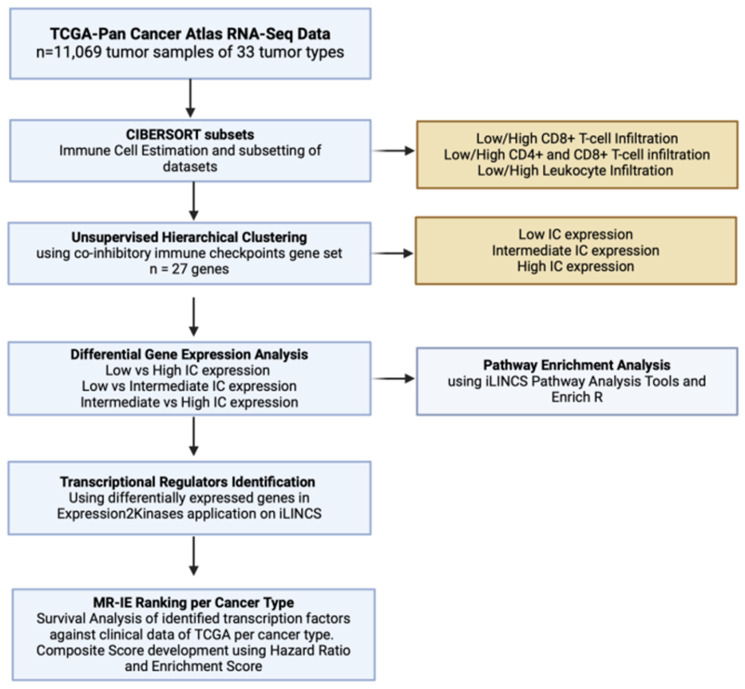
Schematic workflow of integrated in silico analysis to identify MR-IE. Blue boxes represent analyses conducted, and yellow boxes represent details of the analyses. This figure was made with Biorender.com.

**Figure 2 cancers-16-04197-f002:**
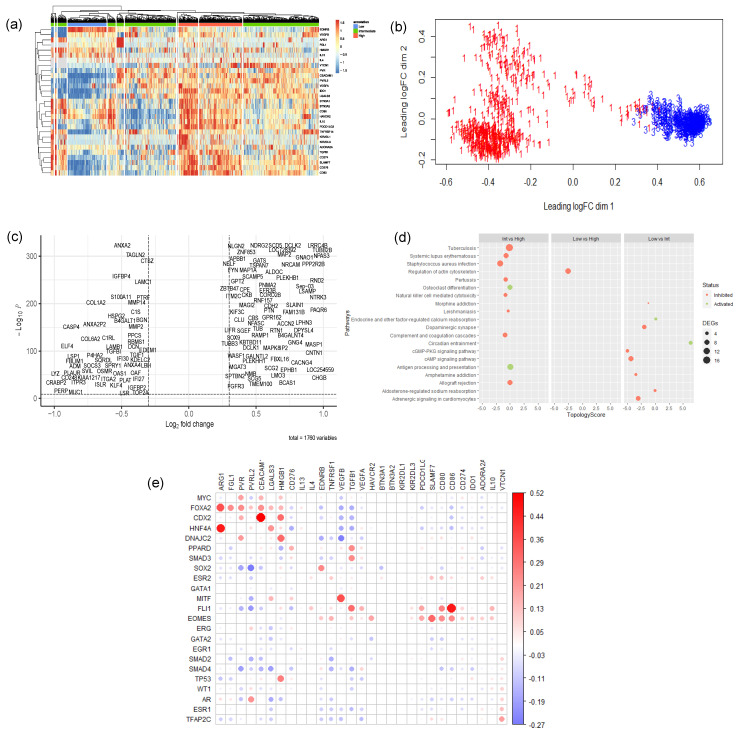
Representative analysis of low CD4+ and CD8+ T-cell estimated TCGA samples. (**a**) Unsupervised hierarchical clustering of TCGA Pan-Cancer samples, a subset with low estimates of CD4+ and CD8+ T-cells against co-inhibitory IC gene expression. (**b**) Multidimensional Scaling Plot of clusters defined at high (red) and low (blue) IC. (**c**) Volcano Plot of differentially expressed genes defined at *p* < 0.001. Red dots denote significantly differentially expressed genes (**d**) Signaling Pathway Impact Analysis of all three analyses. Red dots represent inactivated pathways, and green dots represent activated pathways. (**e**) Correlation Analysis of enriched transcription factors against ICs. Red dots represent a positive correlation, whereas blue dots represent a negative correlation. Empty spaces denote insignificant correlation.

**Figure 3 cancers-16-04197-f003:**
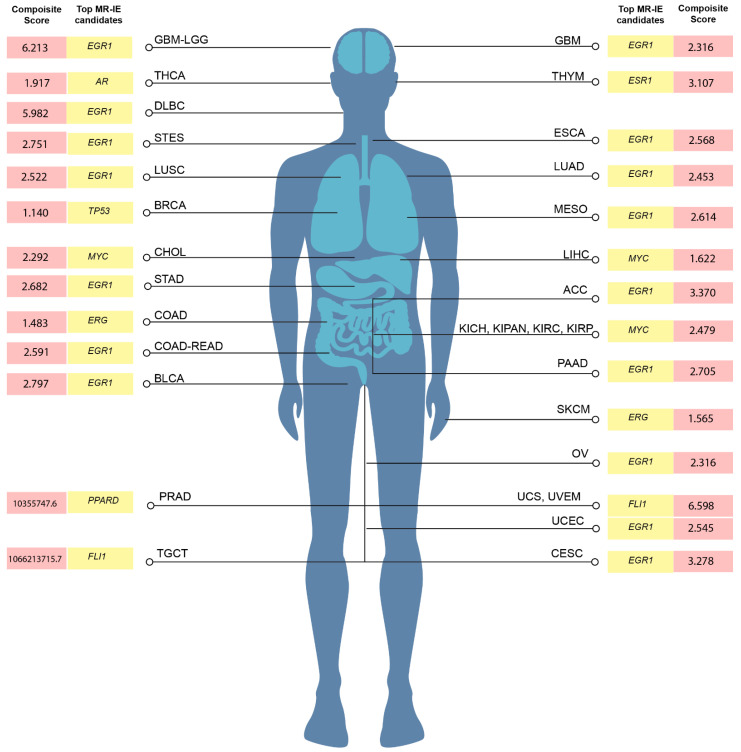
Top-ranking candidate MR-IE for each TCGA tumor type based on the composite score. The composite score was calculated using the Hazard Raito (the effect of the identified transcription factor’s effect on patients’ overall survival in each tumor type) multiplied by the statistical enrichment score obtained from the Expression2Kinases application. The composite score for the candidate MR-IE of KICH, KIRC, and KIRP is represented by KIPAN. The composite score for the candidate MR-IE for UCS, UVEM, is represented by UVEM. Candidates not represented in this figure can be found in Appendix A.

**Figure 4 cancers-16-04197-f004:**
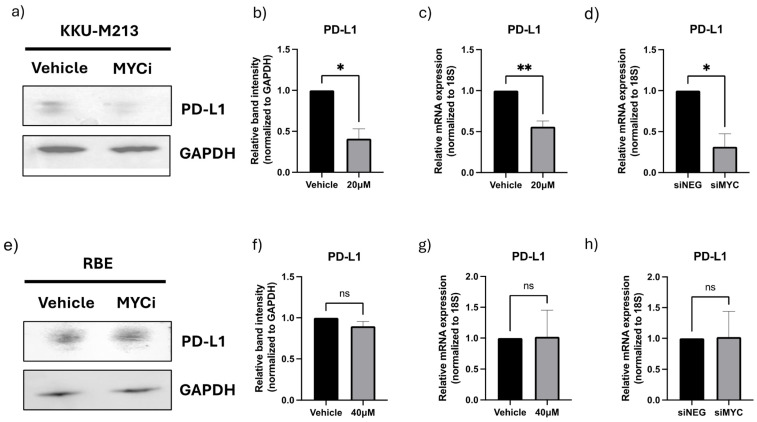
Inhibition or knockdown of master regulator results in the downregulation of PD-L1 expression in CCA cell lines, KKU-213, but not RBE. (**a**) Western Blot of PD-L1 after MYC inhibitor treatment in KKU-213 cells. (**b**) Densitometry analysis of PD-L1 bands normalized to GAPDH in KKU-213. (**c**) The mRNA expression of PD-L1 normalized to 18S expression after MYC inhibitor treatment in KKU-213 cells. (**d**) The mRNA expression of PD-L1 normalized to 18S after MYC siRNA knockdown in KKU-213 cells. (**e**) Western Blot of PD-L1 protein after MYC inhibitor treatment in RBE cells. (**f**) Densitometry analysis of PD-L1 bands normalized to GAPDH in RBE. (**g**) The mRNA expression of PD-L1 normalized to 18S after MYC inhibitor treatment in RBE cells. (**h**) The mRNA expression of PD-L1 normalized to 18S after MYC siRNA knockdown in RBE cells. The statistical analyses and data represent an analysis of a minimum of three experiments conducted. Statistically significant values were taken at *p*-value < 0.05 (*), 0.01 (**), else it is considered ‘non significant’ (ns). The original Western blot figures can be found in Appendix A.

**Figure 5 cancers-16-04197-f005:**
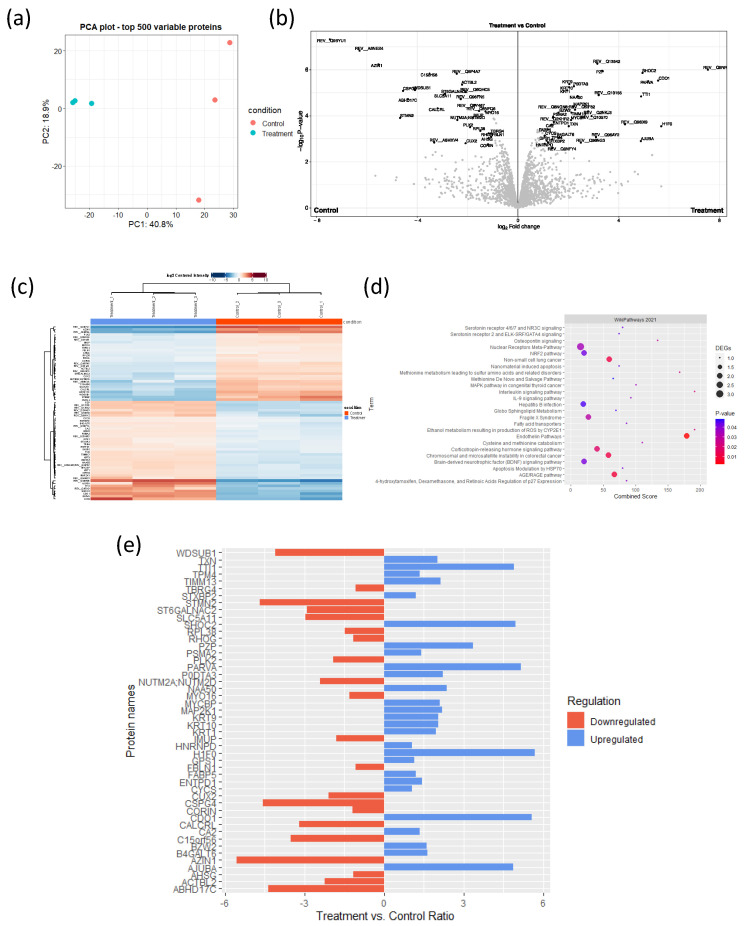
Proteomics analysis reveals the inhibition or knockdown of master regulator results in the modulation of immune-related markers in the cancer cell line proteome in the KKU-213 cell line. (**a**) Principle Component Analysis of the overall expression of identified proteins of KKU213 cell samples, treated either with the MYC inhibitor or Vehicle control. (**b**) Volcano Plot of differentially expressed proteins considered at Log2Fold change of two and *p*-value < 0.05. (**c**) Heatmap of differentially expressed proteins between KKU-213 treated samples and control. (**d**) Pathway enrichment of differentially expressed proteins from the WikiPathways Database. (**e**) Relative expression of immune-related proteins after MYC inhibitor treatment compared to the control.

**Figure 6 cancers-16-04197-f006:**
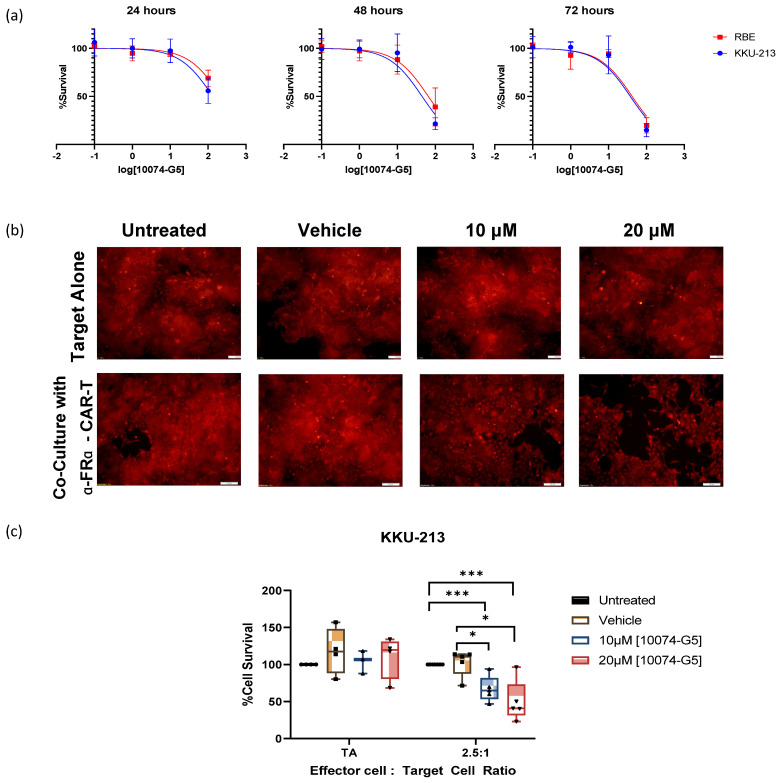
Inhibition of MYC (MR-IE of CCA) results in the potentiation of immune-mediated cell death through CAR-T cell-based therapy in the KKU-213 cell line. (**a**) Dose-response curves of CCA cell lines KKU-213 and RBE, treated with varying doses of MYC inhibitor for 24, 48, and 72 h. The *X*-axis represents log concentrations of the MYC inhibitor (10074-G5) at points 0.1, 1, 10, and 100 µM. (**b**) Fluorescent images (taken at magnification of 10×; Scale bar represents 100 µm) of KKU-213 tagged with mCherry, treated with MYC inhibitor in varying doses and co-cultured with anti-FR-α CAR-T cells. (**c**) Cell survival plot compared between target cell alone (TA), and co-cultured with anti-FR-α at an effector to the target cell ratio of 2.5:1. The statistical analyses and data represented are an analysis of a minimum of three experiments conducted. Statistically significant values were taken at *p*-value < 0.05 (*), 0.001 (***).

## Data Availability

The dataset used in this investigation can be obtained from the Genomic Data Commons Portal of The Cancer Genome Atlas (TCGA) in the Pan-cancer Atlas publication page (https://gdc.cancer.gov/about-data/publications/pancanatlas, accessed on 16 May 2020). The software used is listed in the Key Resources Table.

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
