# Peer review of "Identification of Transcriptional Regulators of Immune Evasion Across Cancers: An Alternative Immunotherapeutic Strategy for Cholangiocarcinoma"

_cancers, 2024, doi:10.3390/cancers16244197_

Round 1
Reviewer 1 Report
Comments and Suggestions for Authors
This study offers a candidate list of MR-IE immunotherapeutic targets per cancer type and also highlights the promise of MR-IE inhibition as a novel potent immu- 61 notherapeutic strategy for the treatment of CCA,providing insight into the regulation of immune evasion as an alternative immunotherapeutic strategy.The article has implications for clinical practice, but the potential synergistic effect rather than simply additive between MYC inhibition and CAR-T cell therapy should be further addressed.Additionally, the study only used a limited number of cell lines to assess the combination of MYC inhibition with CAR-T therapy, which lacks universal.Please discuss the reasons for choosing these cells.
Author Response
We humbly thank the reviewer for their insightful comments and aligned vision about the clinical implications of this investigation. To address the potential synergistic effect of MYC inhibition with CAR-T in CCA, we have included the following lines:
In Line 789-813 we discuss the synergism between treatment strategies.
Most importantly, the modulation of these markers culminates in the functional reinstatement of immune mediated cytotoxicity of CCA cells (Figure 6). To further the translatability of our novel immunotherapeutic concept we sought to combine our therapeutic strategy with CAR-T therapy. Various clinical investigations revealed that CAR-T therapies have seen success in hematological malignancies, but were less efficacious in solid tumors due to its lack of penetrability in tumors [35]. This gave rise to the combination of CAR-T therapies with ICB to sensitize solid cancer cells to CAR-T infiltration and further potentiate immune mediated cell death, such as pembrolizumab and mesothelin-specific CAR-T cells [36]. In a similar vein, our investigation considered the immune cell infiltration and IC expression in the identification of MR-IE candidates, with the aim of sensitizing cancer cells to multiple facets of immune surveillance. Additionally, we combined cancer cell inhibition of MR-IE with fourth-generation CAR-T cells in CCA cells. We observed that there is a synergistic effect of CCA cytotoxicity when compared to MYC inhibition alone and CAR-T cell therapy alone. This potentially suggests that both treatments address the limitations of the other, while being more efficacious in mitigating cancer cells. Prospective studies may investigate whether the combined treatment increases immune cell infiltration of tumors and anti-tumoral inflammation in vivo. Taken together, our preclinical investigation provides confidence in our workflow to identify and rank candidate MR-IEs per cancer type, that can be developed as immunotherapeutic targets.
To address the limitation of choice of cell lines, we have included the following lines in lines 825-829:
Our investigation was limited in the use of fewer cell line representatives. However, the cell lines chosen were based on prior reports of KKU-213 being immunogenic and responsive to cell-based immunotherapies [37,38], and RBE was seldom reported to be immunogenic. This encourages the development of more immunogenic CCA cell lines to further explore various immunotherapeutic strategies.
We hope these amendments adequately address your valuable comments.
Reviewer 2 Report
Comments and Suggestions for Authors
Paper submitted to cancers MDPI, titled: “Identification of transcriptional regulators of immune evasion across cancers: an alternative immunotherapeutic strategy for cholangiocarcinoma” by Simran Venkatraman et al., evaluates master regulators of immune evasion (MR-IE) and immune checkpoint (IC) expression in multiple cancers, and finally sets the attention on closer analysis of cholangiocarcinoma. Additional experiments upon treatment with c-Myc inhibitor/siRNA and CAR-T cell therapy provide direct evidence for c-Myc acting like MR-IE in cholangiocarcinoma, that upon targeting c-Myc with drug, increased response to CAR-T cell immunotherapy is observed.
Due to the confusing bioinformatics data interpretation, and lack of data supporting the FAa-CAR-T therapy, the paper is not recommended for publication, or major revisions are recommended in order to improve the clarity of methods and results presented.
The connection of different sides of cancer biology combined with bioinformatics needs to be properly demonstrated, right now major findings are not well summarized and are not presented with sufficient clarity, but are summarized in cartoonish Figures (Figure3, and Graphical Abstract).
Here are specific critiques:
1. Throughout the paper authors refer to Tables, that are in the supplementary materials, I assume supplementary Tables are intended for the major paper and data discussed directly in the Results section.
2. The parts for bioinformatics determination of hits for MR-IE is confusing and not convincing to the properly used applications and selection criteria for the top hits, presented on Figure 3.
3. Throughout the text there are several misspellings that need to be corrected, and paper needs to focus on the specific MR-IE, with direct explanation how the master regulators were determined and based on what criteria, and models supporting involvement of any given MR-IE, whether it is based on the gene expression analysis or chances for longer/shorter survival of patients with a given gene up-regulated.
4. If patient survival is one of the criteria taken for selection of MR-IE, please provide such data in the Figure or supplementary material.
5. Authors do not delve into the construction of CAR-T cell therapy and yet in the Graphical Abstract and in the Methods mention production of CAR-T cells. If CAR-T cell therapy was used in this paper it needs to be in detail provided the description of the construct carrying the chimeric receptor, its source and activating domains and in depth characterization of the sensitivity of cancer cells to given therapy as a single agent. In addition, evaluation in vitro of the antigen on the selected cancer cell line, would help understand the CAR-T cell therapy mechanism of action. This is especially significant for the increased tumor response to immunotherapy, as observed in Figure 6.
6. In Methods section, Cell Viability and Proliferation Assay, authors forget to mention where the c-Myc inhibitor was purchased from, and the sequence for siRNA targeting c-Myc is also omitted.
7. In the Methods section, Immune Mediated Cell Death authors talk about fluorescent assay based, presented in Figure 6, and absorption based, for RBE cells stained with Crystal Violet, method that is irrelevant as data are not presented in this paper.
8. siRNA experiments of c-Myc, or c-Myc inhibitor experiments although convincing in the data provided, still would need additional in depth description of Method used and final end result connection. For example, why there is c-Myc inhibitor combination with CAR-T cell therapy and cancer hypersensitivity of one cell line, KKU-213, but the data with siRNA, complementary to the same type of experiment are not included? Is siRNA targeting c-Myc didn’t result in hypersensitivity to CAR-T cell therapy? Please comment this discrepancy.
Minor critiques that also prove that the paper is not ready for publication in current state, are following:
1. Figure 2 lacks description of the Legend of Figure 2 D, E, F.
2. In Supplementary data there is no oligo listing for siRNA used for c-Myc knockdown experiments.
Comments on the Quality of English Language
English language is not the biggest problem, but can be improved. Understanding of the topic, and proper use of the control experiments, data visualization and shortening of the bioinformatics analysis might help clarify the content.
Author Response
Author’s response:
We are grateful for the reviewer’s dedication towards a thorough review of our investigation. Taking your points into consideration, we have amended the manuscript as follows:
- All references to Tables have been amended to ‘Supplementary Table XX’
- To clarify the process of identifying the MR-Ies per cancer type, the analytical pipeline in Lines 392-415 was revised, as the following:
“Immune evasion occurs through the synchronous process of immune cell infiltration, co-inhibitory IC expression and modulation of immune related markers. Thus, an ideal therapeutic target would be a transcriptional regulator that governs all three processes. To identify transcriptional regulators that controls these processes, we must obtain all the genes that are affected in varying degrees of estimated immune cell infiltration, and varying degrees of inhibitory IC expression. To do so, the PanCancer TCGA samples must be subset according to varying degrees of estimated infiltration and then stratified into varying degrees of co-inhibitory IC expression. The transcriptome of these data subsets and clusters are then compared with each other to unravel the different signaling pathways associated with the immune evasion process. As the resulting DEGs reflect all genes associated with the process of immune evasion, The DEGs from these comparisons are then used as an input for transcription factor enrichment. This analysis helps to identify transcription factors that bind to common motifs identified amongst the DEGs. Thus, this yields candidate MR-IEs from varying degrees of leukocyte estimates and IC expression across cancers. We acknowledge that not all cancers are driven by the same targets, in fact, some therapeutic targets may have opposite roles in different cancers. To overcome this issue, we employed survival analysis to assess the effect of the gene expression of these transcription factor on patient overall survival per cancer type. Using the Hazard Ratio provided from this analysis, and the enrichment score of the transcription factor enrichment analysis, a composite score was developed to rank these candidate MR-IEs per cancer type. Together, this bioinformatics workflow identifies and ranks transcriptional regulators of immune evasion that affects genes involved in varying degrees of immune infiltration, and varying degrees of IC expression, per cancer type. This workflow is summarized in Figure 1.”
Additionally, the methods have been updated with two additional sections, as follows:
Survival Analysis
Survival analysis was performed using the survival, survminer, and survplot packages
in R version 4.0.2. Cox-Proportional Univariate Regression model was applied to each significantly enriched transcription factor against the survival data of each TCGA cancer type. This script was adapted from Mikhail Dozmorov’s analysis script found in (https://github.com/mdozmorov/TCGAsurvival?tab=readme-ov-file#readme). Hazard Ratios (HR) and Wald Statistic p-value were tabulated.
Composite Scoring of Transcription Factors
A composite score was calculated by multiplying the Average Enrichment Score provided by Expression2Kinases for each significantly enriched (p-value < 0.05) transcription factor, and the Hazard Ratio for each transcription factor against each cancer type. This was used to rank the top MR-IE candidates per cancer type.
To clarify on the selection criteria of the top hits listed in Figure 3, we added the following lines to the results:
Additionally, Figure 3 has been amended to include the respective composite scores.
- We hope the revisions mentioned in (2) adequately addresses the tools and models chosen to rank and identify the master regulators of immune evasion per cancer type. We have also amended the results section to include these lines:
Line 483- 486: Using the DEGs obtained from the aforementioned comparisons across different estimated immune cell infiltration and different IC expression levels, we sought to identify common transcriptional regulators that bind to the common motifs amongst these DEGs, using Expression2Kinases.
Line 498-507: The identified transcriptional regulators across all cancer types are statistically scored. However, different transcriptional regulators may have varying effects on patient mor-tality. Thus, we conducted a cox-proportionate univariate regression analysis of each transcriptional regulator for each of the 33 TCGA cancer types, to yield a hazard ratio. Using this hazard ratio multiplied by the enrichment score allowed us to rank the significant transcription factors per cancer type. The hazard ratio, enrichment scores, and the combined Composite Score of candidate MR-IE per cancer type are tabulated in Sup-plementary Table 2. The top-ranking candidate MR-IE per cancer type is depicted in Figure 3, Supplementary Table 2 and 3. The cancer types are listed as per TCGA study abbreviations.
- We used Overall Survival to calculate Hazard Ratios of each enriched transcription factor for each TCGA cancer type. This data, the enrichment score and p-value, as well as the calculated Composite Score is already listed in Supplementary Table 2.
- Details of the development of CAR-T cells has been improved and referred to from a prior publication that established and characterized the fourth generation CAR-T cell intended for breast cancer use. This is now revised in lines 353-359. It reads:
“The fourth generation anti-FRÉ‘-CAR-T cells has been established by Luangwattananun et al., indicated towards breast cancer cell lines. Characterization and details of its construct has been established in their publication [17]. Briefly, the construct contains a single specific chain variable fragment targeting FRÉ‘. In the fourth-generation iteration of this CAR-T cell, three intracellular costimulatory domains (CD28, 4-1BB, and CD27) linked to CD3ζ, was added to the construct. Our investigation repurposed this CAR-T cell towards CCA cells expressing FRÉ‘.”
Additionally, the evaluation of expression of FRÉ‘ in CCA cell lines including the selected cell line was explained in Lines 667-670, referring to Supplementary Figure 11.
The effect of the CAR-T cell alone on KKU-213 cell line is listed as a part of Untreated control when combined with CAR-T cells. However, we appreciate the point of the reviewer in wanting to emphasize that the CAR-T indeed is functional in the targeted cytotoxicity of CCA cells.
- The sequence of the siRNA for MYC was not revealed as it was purchased from SantaCruz, who also haven’t revealed their sequence in the datasheet. Details of the siRNA purchased, is now reflected in Lines 289-290. “siRNA of MYC (human) (sc-29226) was purchased from SantaCruz Biotechnology (SantaCruz, Biotechnology, TX, USA)”. Information regarding where the MYC inhibitor (10074-G5) was purchased is now listed under the Cell Viability and Proliferation Assay section (line 332). “…treated with either vehicle control or MYC inhibitor (10074-G5, purchased from MedChemExpress, NJ, USA)”.
- We thank the reviewer for their suggestion for relevancy and specificity. Respectfully, while the results of RBE is not presented in the main figures albeit the Supplementary Figure 12, we would like to be transparent in how we obtained the results for both cell lines.
- We have updated the discussion to address the discrepancy in experiments using siRNA knockdown to highlight changes in markers and functional assays to highlight immune mediated cell death in the lines 829-836:
The present investigation also attempted to explore the use of siRNA knockdown as a form of therapy, however this did not yield any significant changes to cell viability in CCA cells alone nor with co-culture. While the knockdown of MYC was transient enough to observe transcriptional regulation, the duration was not sufficient to last during co-culture with CAR-T therapy (data not shown). Additionally, the transient knockdown may eventually lead to the compensation of protein production, thus nullifying its efficacy in CCA cytotoxicity. Thus, this investigation utilized the knockdown by siRNA treatment to highlight modulations of key markers.
Amendments for Minor Critiques:
- We thank the reviewer for bringing this to our attention. We have ameneded Figure 2 figure legends which was unwittingly cut off, as follows:
“Figure 2 – Representative analysis of low CD4+ and CD8+ T-cell estimated TCGA samples. a) Unsupervised hierarchical clustering of TCGA Pan Cancer samples subset with low estimates of CD4+ and CD8+ T-cells against co-inhibitory IC gene expression. b) Multidimensional Scaling plot of clusters defined at high (red) and low (blue) IC expression. c) Volcano Plot of differentially expressed genes defined at p<0.001. Red dots denote significantly differentially expressed genes d) Signaling Pathway Impact Analysis of all three analyses. Red dots represent inactivated pathways and green dots represent activated pathways. e) Correlation Analysis of enriched transcription factors against ICs. Red dots represent positive correlation, whereas blue dots represent negative correlation. Empty spaces denote insignificant correlation.”
- The oligo used for siRNA knockdown of MYC could not be provided as the siRNA mix for MYC was purchased from SantaCruz who have not revealed the sequence for commercial purposes. We have reflected the same in the Methods as per point 6 of your review.
We hope these amendments adequately address your valuable comments, and we thank the reviewer for guiding us to improve the quality of this manuscript.
Reviewer 3 Report
Comments and Suggestions for Authors
The study demonstrates investigated regulators of immune evasion across cancer types. The topic is of great interest, while extensive revision is need.
1. The abstract is quite long and unclear what authors would like to focus on. Please consider to re-write the abstract without any sectioning as Backgroud, Methods, Results, and Conclusions, and to be consise.
2. Re-formatting of the manuscript into the Journal standard is needed. Background needs to be changed into Introduction, and description on transcriptional regulators of ICs need to be expanded. Please spell out the acronyms at the first time shown.
3. Methods section needs to be revised to describe identification of transcriptional regulators more in detail. Production of the fourth generation of FRalpha-CAR-T cells is not clear enough to demonstrate what "the fourth generation" means. Is it the passage number of the cell culture?
4. Figure 2 is not clear enough. Volcano Plot of differentially expressed genes defined at p. in lines 272-273 is quite vague. The legends for Figure 2d and e are needed.
5. Figure 3 shows Composite Score calculated using the Hazard Ratio in the legend, while no value is indicated in the figure. Please indicate the composite score inside the figure.
6. Figure 5 needs to be revised with clearer pictures and detailed legends.
7. Figure 6 needs higher resolution. Detailed explanation of X axis of Figure 6a is needed.
8. Please add some more discussions and references.
9. Conclusion needs to be revised to emphasize the finding of the study.
Author Response
Author’s response: We sincerely thank the reviewer for their detail-oriented review to help the quality of this manuscript. We have amended the following:
- Abstract has been re-written for brevity and specificity. Abstract sections have been removed.
- Background has been corrected to ‘Introduction’ in Line 174. Introduction to transcriptional regulators of ICs have been expanded in the introduction in Lines (160 onwards). Abbreviations have been spelled out the first time it is used in the main text.
- Details of the generation of CAR-T cells has been improved in lines 353-359. It now reads: The fourth generation anti-FRÉ‘-CAR-T cells has been established by Luangwattananunet al., indicated towards breast cancer cell lines. Characterization and details of its construct has been established in their publication [17]. Briefly, the construct contains a single specific chain variable fragment targeting FRÉ‘. In the fourth-generation iteration of this CAR-T cell, three intracellular costimulatory domains (CD28, 4-1BB, and CD27) linked to CD3ζ, was added to the construct. Our investigation repurposed this CAR-T cell towards CCA cells expressing FRÉ‘.
- The figure and Figure legend have been amended
- The Figure has been re-inserted with a higher resolution.
- The figure legend now reads:
Figure 2 – Representative analysis of low CD4+ and CD8+ T-cell estimated TCGA samples. a) Unsupervised hierarchical clustering of TCGA Pan Cancer samples subset with low estimates of CD4+ and CD8+ T-cells against co-inhibitory IC gene expression. b) Multidimensional Scaling plot of clusters defined at high (red) and low (blue) IC expression. c) Volcano Plot of differentially expressed genes defined at p<0.001. Red dots denote significantly differentially expressed genes d) Signaling Pathway Impact Analysis of all three analyses. Red dots represent inactivated pathways and green dots represent activated pathways. e) Correlation Analysis of enriched transcription factors against ICs. Red dots represent positive correlation, whereas blue dots represent negative correlation. Empty spaces denote insignificant correlation.
- Figure 3 is now amended to include the respective MR-IE composite score per cancer type. The corresponding figure legend reads:
- Figure 3. – Top-ranking candidate MR-IE for each TCGA tumor type based on composite score. Composite Score was calculated using the Hazard Raito (effect of identified transcription factor’s effect on patients’ overall survival in each tumor type) multiplied by the statistical enrichment score obtained from the Expression2Kinases application. Composite Score for the candidate MR-IE of KICH, KIRC, KIRP, is represented by KIPAN. Composite Score for candidate MR-IE for UCS, UVEM is represented by UVEM. Candidates not represented in this figure can be found in Supplementary Table 2.
- A higher resolution image of Figure 5 has been appended. The figure legends have been rewritten with more specific details:
Figure 5 – Mass spectrometry analysis reveals inhibition or knockdown of master regulator results in modulation of immune related markers in the cancer cell line proteome in KKU-213 cell line. a) Principal Component Analysis of the overall expression of identified proteins of KKU213 cell samples either treated with MYC inhibitor or Vehicle control. b) Volcano Plot of differentially expressed proteins considered at log2fold change of 2 and p.value < 0.05. c) Heatmap of Differentially expressed proteins between KKU-213 treated samples and control. d) Pathway Enrichment of differentially expressed proteins from WikiPathways Database. e) Relative expression of Immune related proteins after MYC inhibitor treatment compared to control.
- A higher resolution image of Figure 6 is appended. The figure legend is rewritten to explain the axes, as follows:
Figure 6 - Inhibition or MYC (MR-IE of CCA) results in the potentiation of immune-mediated cell death through CAR-T cell-based therapy in KKU-213 cell line. a) Dose-response curves of CCA cell lines KKU-213 and RBE treated with varying doses of MYC inhibitor for 24, 48, and 72 hours. X-axis represents log-concentrations of the MYC inhibitor (10074-G5) at points 0.1, 1, 10, 100µM. b) Fluorescent images of KKU-213 tagged with mCherry treated with MYC inhibitor in varying doses and co-cultured with anti-FR-É‘ CAR-T cells. c) Cell Survival plot compared between target cell alone (TA), and co-culture with anti-FR-É‘ at an effector to target cell ratio of 2.5:1. Statistical analyses and data represented is an analysis of a minimum of 3 experiments conducted.
- We thank the reviewer for their suggestion to improve the discussion with more references. Accordingly, in Line 789-806 we discuss the synergism between treatment strategies.
Most importantly, the modulation of these markers culminates in the functional reinstatement of immune mediated cytotoxicity of CCA cells (Figure 6). To further the translatability of our novel immunotherapeutic concept we sought to combine our therapeutic strategy with CAR-T therapy. Various clinical investigations revealed that CAR-T therapies have seen success in hematological malignancies, but were less efficacious in solid tumors due to its lack of penetrability in tumors [35]. This gave rise to the combination of CAR-T therapies with ICB to sensitize solid cancer cells to CAR-T infiltration and further potentiate immune mediated cell death, such as pembrolizumab and mesothelin-specific CAR-T cells [36]. In a similar vein, our investigation considered the immune cell infiltration and IC expression in the identification of MR-IE candidates, with the aim of sensitizing cancer cells to multiple facets of immune surveillance. Additionally, we combined cancer cell inhibition of MR-IE with fourth-generation CAR-T cells in CCA cells. We observed that there is a synergistic effect of CCA cytotoxicity when compared to MYC inhibition alone and CAR-T cell therapy alone. This potentially suggests that both treatments address the limitations of the other, while being more efficacious in mitigating cancer cells. Prospective studies may investigate whether the combined treatment increases immune cell infiltration of tumors and anti-tumoral inflammation in vivo. Taken together, our preclinical investigation provides confidence in our workflow to identify and rank candidate MR-IEs per cancer type, that can be developed as immunotherapeutic targets.
We further discussed the limitations of the investigation in the lines 825 to 836:
Our investigation was limited in the use of fewer cell line representatives. However, the cell lines chosen were based on prior reports of KKU-213 being immunogenic and responsive to cell-based immunotherapies [37,38], and RBE was seldom reported to be immunogenic. This encourages the development of more immunogenic CCA cell lines to further explore various immunotherapeutic strategies. The present investigation also attempted to explore the use of siRNA knockdown as a form of therapy, however this did not yield any significant changes to cell viability in CCA cells alone nor with co-culture. While the knockdown of MYC was transient enough to observe transcriptional regulation, the duration was not sufficient to last during co-culture with CAR-T therapy (data not shown). Additionally, the transient knockdown may eventually lead to the compensation of protein production, thus nullifying its efficacy in CCA cytotoxicity. Thus, this investigation utilized the knockdown by siRNA treatment to highlight modulations of key markers.
- The conclusion has been made into a separate section and is re-written in lines 839-851:
Conclusion
Our investigation collectively introduces transcriptional regulators as alternative immunotherapeutic targets for the development of cancer treatments. We also provided insight into the transcriptomic changes associated with immune evasion and its regulation. Moreover, our investigation provides a robust bioinformatics workflow that ranks and identifies MR-IE per cancer type. As a result, MYC was identified to be the candidate MR-IE for CCA, that downregulates key immune related markers such as PD-L1 expression, IFNA-21, HLA-B and IL-18. These modulations resulted in CCA cells being sensitized to CAR-T mediated cell death. Conclusively, we encourage the development of several small-molecule inhibitors against MYC for the treatment of CCA. We hope the findings of this investigation inspires further development of these MR-IEs as therapeutic targets to benefit a wider patient population.
We hope this revisions adequately addresses the reviewer’s valuable comments to help improve this manuscript.
Reviewer 4 Report
Comments and Suggestions for Authors
One of the hallmarks of cancer cells is immune evasion. Thus, the development of a new anticancer strategy targeting immune system response is highly appreciated. The manuscript by Venkatraman et al. aims to introduce a novel and effective immunotherapeutic strategy by targeting master regulators of immune evasion (MR-IE) and offer more potent immunotherapeutic treatment options for cholangiocarcinoma (CCA).
The research is based on database analyses and in vitro experiments in CCA cell lines. Top-ranking candidate MR-IE for 33 tumor types was presented based on The Cancer Genome Atlas. Moreover, MYC was identified as the high-ranking candidate MR-IE for CCA.
The manuscript is interesting and important for readers working in the field of immunotherapy and gastrointestinal tumors.
Minor issues:
1. In the Cancers journal, a Simple Summary before the abstract is needed; the word limit is 150.
2. The abstract should not contain headings like Background, Methods, etc. It should also be shortened to a maximum of 200 words.
3. In the main text, all abbreviations should be explained when the first time is mentioned.
4. Figure 2 - description of part d) and e) is missing.
5. For experimental results (Figures 4, 6), please add information about a number of experiments (n=?).
6. Line 426 - "(Table 6)" should be probably Figure 6.
Author Response
Author’s Response: We thank the reviewer for alerting us to the journal’s format expectations and of the error in our captions . Accordingly, we have amended the following:
- A Simple Summary has been added before the abstract (Lines 37-47).
- Abstract has been re-written for brevity and specificity. It now has 198 words. (Lines 93-108).
- Abbreviations have been spelled out on first use, wherever applicable.
- The figure legend has been completed, it now reads:
- Figure 2 – Representative analysis of low CD4+ and CD8+ T-cell estimated TCGA samples. a) Unsupervised hierarchical clustering of TCGA Pan Cancer samples subset with low estimates of CD4+ and CD8+ T-cells against co-inhibitory IC gene expression. b) Multidimensional Scaling plot of clusters defined at high (red) and low (blue) IC expression. c) Volcano Plot of differentially expressed genes defined at p<0.001. Red dots denote significantly differentially expressed genes d) Signaling Pathway Impact Analysis of all three analyses. Red dots represent inactivated pathways and green dots represent activated pathways. e) Correlation Analysis of enriched transcription factors against ICs. Red dots represent positive correlation, whereas blue dots represent negative correlation. Empty spaces denote insignificant correlation.
- The following statement has been added in the figure legend to declare number of experiments conducted.
- “Statistical analyses and data represented is an analyses of a minimum of 3 experimentsconducted”
- We have amended the text to reflect ‘Figure 6’ as well as ‘Supplementary Table 6’ which refers to the IC50s of the MYC inhibitor against the CCA cell lines alone.
We hope these improvements adequately address the reviewer’s valuable comments in improving this manuscript.
Round 2
Reviewer 2 Report
Comments and Suggestions for Authors
Authors corrected manuscript and included more clarifying text on CAR-T cells production. Overall it is of interest to publish this manuscript, since it presents a cohesive story.
It is advised though to edit carefully this manuscript prior publication for any errors or misspellings or other grammar mistakes.
Author Response
We thank the reviewer for their generosity in providing more time, and for giving us a thorough constructive review on our work.
Reviewer 3 Report
Comments and Suggestions for Authors
The authors addressed the reviewer's comments.
Author Response
We thank the reviewer for their constructive feedback.